# Role of Single Nucleotide Variants in *FSHR*, *GNRHR*, *ESR2* and *LHCGR* Genes in Adolescents with Polycystic Ovary Syndrome

**DOI:** 10.3390/diagnostics11122327

**Published:** 2021-12-11

**Authors:** Lasma Lidaka, Laine Bekere, Adele Rota, Jekaterina Isakova, Gunta Lazdane, Anda Kivite-Urtane, Iveta Dzivite-Krisane, Inga Kempa, Zane Dobele, Linda Gailite

**Affiliations:** 1Department of Paediatric Gynaecology, Children’s Clinical University Hospital, LV-1004 Riga, Latvia; 2Department of Obstetrics and Gynaecology, Riga Stradins University, LV-1007 Riga, Latvia; laine.bekere@gmail.com; 3Scientific Laboratory of Molecular Genetics, Riga Stradins University, LV-1007 Riga, Latvia; dr.adele.rota@gmail.com (A.R.); jekaterina.isakova.25@gmail.com (J.I.); inga.kempa@rsu.lv (I.K.); zane.dobele@rsu.lv (Z.D.); linda.gailite@rsu.lv (L.G.); 4Institute of Public Health, Riga Stradins University, LV-1007 Riga, Latvia; gunta.lazdane@rsu.lv (G.L.); anda.kivite-urtane@rsu.lv (A.K.-U.); 5Department of Paediatric Endocrinology, Children’s Clinical University Hospital, LV-1004 Riga, Latvia; dzivite@bkus.lv

**Keywords:** adolescents, genetics, *GNRHR*, *ESR2*, *LHCGR*, *FSHR*, polycystic ovary syndrome

## Abstract

Background: Polycystic ovary syndrome (PCOS) is the most common endocrinopathy in women, affecting up to 16.6% of reproductive-age women. PCOS symptoms in adolescents comprise oligomenorrhoea/amenorrhoea and biochemical and/or clinical hyperandrogenism. Long-term health risks of PCOS patients include infertility, metabolic syndrome, type 2 diabetes and cardiovascular disease. Genetic factors have been proven to play a role in development of the syndrome and its symptoms. Objective: To investigate single nucleotide variants (SNVs) in the *GNRHR, ESR2, LHCGR* and *FSHR* genes in adolescent patients with PCOS and their association with PCOS symptoms. Methods: We conducted a cross-sectional study comprising of 152 adolescents: 63 patients with PCOS, 22 patients at risk of developing PCOS and 67 healthy controls. Participants were recruited from out-patients attending a gynaecologist at the Children’s Clinical University Hospital, Riga, Latvia, between January 2017 and December 2020. Genomic DNA was extracted from whole blood, and SNVs in the *GNRHR, ESR2, LHCGR* and *FSHR* genes were genotyped. The distributions of SNV genotypes were compared among the three groups and genotype-phenotype associations within the PCOS group were evaluated. Results: No statistically significant differences were found in the distributions of genotypes for *GNRHR* (rs104893837), *ESR2* (rs4986938), *LHCGR* (rs2293275) and *FSHR* (rs6166, rs6165, rs2349415) among PCOS patients, risk patients and healthy controls. Within the PCOS group, *ESR2* rs4986938 minor allele homozygous patients had a significantly higher level of total testosterone than major allele homozygous patients and heterozygous patients. A significantly higher total testosterone level was also observed in PCOS patients carrying the *LHCGR* rs2293275 minor allele compared with major allele homozygous patients. Conclusions: The SNVs *ESR2* rs4986938 and *LHCGR* rs2293275 play a role in the phenotypic characteristics of PCOS. To fully uncover their influence on the development of PCOS and its symptoms, further studies of larger cohorts and a follow up of this study sample through to adulthood are required. Furthermore, studies of adolescent PCOS patients conducted prior to the latest European Society of Human Reproduction and Embryology (ESHRE) criteria (2018) should be re-evaluated as the study groups might include risk patients according to these updated criteria, thereby potentially significantly impacting the published results.

## 1. Introduction

Polycystic ovary syndrome (PCOS) is the most common endocrinopathy in women, affecting more than 15% of reproductive-age women. In the adolescent population, it is characterised by oligo/anovulation and biochemical and/or clinical hyperandrogenism [1]. As PCOS most often presents during adolescence, it is commonly masked by symptoms of normal puberty, e.g., acne and menstrual irregularities. The main treatment to target pathophysiological mechanisms of PCOS and to improve later prognosis, is lifestyle changes. Therefore, a timely diagnosis via genetic testing is crucial to avoid long-term consequences of the syndrome [1].

Family clustering [2,3], genome-wide association studies (GWAS) [4] and research examining the role of environmental factors have established that PCOS is a multifactorial disease [5]. The genetics of PCOS appear to be very complex. Hypothalamic-hypophysis-ovarian (HHO) axis dysfunction is a pivotal mechanism in the development of the syndrome. Signal transduction in the main regulatory pathway is executed via the gonadotropin-releasing hormone receptor (GNRHR) and steroid hormone receptors. To date, 19 PCOS susceptibility loci have been identified in various GWAS, with four of them located near genes involved in the HHO axis [4]. It might be expected that these GWAS would provide a precise answer regarding the possible genetic cause of PCOS and genotype-phenotype interactions; however, their results are diverse [4]. An explanation for this disparity could be that the robust analyses used in GWAS may overlook genes that have a weak association with the syndrome and rare genetic variants that cause PCOS and influence its symptoms [4]. Therefore, at present, the GWAS available in the literature merely offer contradictory and inconclusive results regarding the association of different single nucleotide variants (SNVs) with the development of PCOS and its related symptoms.

To help address this discordance related to PCOS genetic association studies, we focused our research on SNVs in genes encoding proteins important in the HHO axis, namely *GNRHR* (rs104893837), oestrogen receptor 2 (*ESR2* rs4986938), luteinising hormone/choriogonadotropin receptor (*LHCGR* rs2293275) and follicle-stimulating hormone receptor (*FSHR* rs6166, rs6165, rs2349415). The SNVs were selected based on previous GWAS or other relevant studies. In the case of *GNRHR*, it was selected because it was observed most frequently in the Latvian population [unpublished results]. To the best of our knowledge, only one study investigating genetic polymorphisms in adolescents with PCOS has been published to date. This study by Unsal et al. examined several genes, but only rs6166 and rs6165 SNVs in FSHR are in common with our selected SNVs [6].

## 2. Materials and Methods

### 2.1. Study Population

The study sample of this cross-sectional study consisted of 152 adolescents: 63 patients with PCOS, 22 patients at risk of developing PCOS (see below) and 67 healthy controls. All participants were of European ancestry and attended a single paediatric gynaecologist at the Children’s Clinical University Hospital, Riga, Latvia, at least a year after menarche. The Children’s Clinical University Hospital is the only specialized children’s hospital in Latvia, providing a whole spectrum of services—both in-patient and out-patient—designed specifically for children up to 18 years of age. All services are financed by the state budget and gynaecologists are accessible without a referral from general practitioners or other specialists. This allows utilization of services by patients from different regions and socioeconomic backgrounds. There is no national registry for PCOS patients, but considering the facility for patient recruitment, we could be sure that the study sample consisted of a homogenous Caucasian population, encapsulating the whole population of the country. Recruitment took place between 1 January 2017 and 31 December 2020. A diagnosis of PCOS was established according to the European Society of Human Reproduction and Embryology (ESHRE) 2018 criteria: biochemical (defined as an elevated total testosterone (T) level above the normal laboratory range according to Tanner staging) and/or clinical hyperandrogenism (modified Ferriman-Gallwey (mFG) score ≥ 4) and oligomenorrhoea (defined as a menstrual cycle length more than 45 days at least a year after menarche or more than 35 days three years after menarche) [1]. Healthy adolescents who attended the same gynaecologist for other issues (such as contraception counselling, etc.) were included in the study as a control group. Additionally, patients with hirsutism but who did not fulfil all the PCOS criteria and did not have any exclusion criteria were included in the so-called ‘risk’ group [1].

Patients with other known gynaecological and endocrinological disorders (Cushing’s syndrome, hypothyroidism, endocrine tumours, diabetes, etc.) or other serious comorbidities (e.g., severe liver or kidney disease) and patients who had used hormonal medication within the previous six months were excluded from the study. The study was approved by the Central Medical Ethics Committee of Latvia (protocol no: 1/16-04-12) and followed the guidelines of the Declaration of Helsinki. All participants gave their signed informed consent. Permission was also received from legal guardians for those participants under the age of 16.

### 2.2. Patient Examination

Clinical examination of participants included body mass index (BMI), waist-hip ratio, mFG score, Global Acne Grading System (GAGS) score and information about the characteristics of their menstrual cycle. The GAGS score was interpreted as follows: 0 = no acne; 1–18 = mild acne; 19–30 = moderate acne; 31–38 = severe acne; ≥39 = very severe acne [7]. World Health Organization AnthroPlus software was used to calculate BMI (kg/m^2^) and its percentile, according to age and normal range for adolescent girls [8]. Weights and heights were measured using standardised calibrated measuring devices.

Pelvic ultrasound was performed by a single examiner using either an HD11 XE or Logiq P5 ultrasound machine (Philips, Amsterdam, Netherlands; General Electric, Boston, MA, USA, respectively). Polycystic ovaries were defined as an ovarian volume >10 mL in at least one ovary and no corpus luteum, dominant follicle or other cystic structure in any ovary. Ovarian volume was calculated using the simplified formula for a prolate ellipsoid. The larger ovary was used to evaluate ovarian size.

For PCOS patients, T, androstenedione (A2) and DHEA-SO4 were tested on the 3rd to 5th day of the menstrual cycle. For patients with amenorrhea, testing was performed in any day in the morning. In order to exclude accidental testing during the periovulatory or luteal phase, if menstruation occurred during the next 21 days, repeated testing on the 3rd to 5th day of the menstrual cycle was performed. Additionally, thyroid-stimulating hormone, prolactin, luteinising hormone, follicle-stimulating hormone and oestradiol were tested on the same day of the menstrual cycle in order to exclude other causes of oligomenorrhoea. Androgen levels, including T, were tested in a certified laboratory using an electrochemiluminescence method (Cobas 6000 immunological analyser; Roche, Basel, Switzerland).

### 2.3. Genetic Testing

Peripheral venous blood samples were collected from the study participants into EDTA-containing tubes, and genomic DNA was isolated using Analytik Jena’s innuPREP Blood DNA Mini Kit (Jena, Germany).

Genotyping for rs6166 (*FSHR*) and rs4986938 (*ESR2*) was performed using a multiplex PCR assay GENTERF (developed at Riga Stradins University); rs104893837 (*GNRHR*) was detected by Sanger sequencing using a BigDye Terminator Kit following the manufacturer’s protocol and employing previously described probes (https://www.academic.oup.com/jcem/article/87/4/1607/2374953. Accessed on 1 January 2021); rs2293275 (*LHCGR*) and rs6165 (*FSHR*) were identified with PCR-RFLP methods adapted from previous publications (doi:10.1111/j.1365-2265.2012.04372.x, https://www.pubmed.ncbi.nlm.nih.gov/19387820/. Accessed on 1 January 2021); rs2349415 (*FSHR*) was detected by PCR-RFLP using the restriction enzyme MboI (primers are available upon request).

Duplicates were used to assess the genotyping quality, and for each variant detected by GENTERF or PCR-RFLP, at least 10% of the samples were validated with Sanger sequencing following the manufacturer’s protocol.

To exclude random error and bias in the gene-disease associations, we checked our genotype frequencies were in Hardy–Weinberg equilibrium.

### 2.4. Statistical Methods

Statistical Package for the Social Sciences software (SPSS22.0) was used for all the statistical analyses. Medians were used to describe the central tendency as the data were not normally distributed (Kolmogorov-Smirnov test *p* < 0.05). Parametric (Student’s *t*-test, ANOVA) and nonparametric (Mann-Whitney U test, Kruskal-Wallis test) tests were used for range and interval scale data, and Pearson’s chi-squared test and Fisher’s Exact test were used for nominal scale data. Two-tailed *p* < 0.05 was considered statistically significant.

## 3. Results

The baseline clinical characteristics of the PCOS, risk and control groups are presented in Table 1. To characterise the age of participants by group, we chose a more precise characteristic of the adolescent population—gynaecological age (years since menarche)—which did not significantly differ among the groups (*p* = 0.441). Significantly more adolescents in the PCOS group and risk group showed markers of poor metabolic health. Specifically, a higher BMI, higher waist-hip ratio and higher percentage of participants with obesity were observed. BMI and waist-hip ratio were significantly higher in PCOS and risk group patients than in control group patients, accordingly *p* < 0.001 and *p* = 0.001. In addition, BMI above the 85th percentile was significantly more common in individuals from the PCOS group than from the control and risk groups (*p* < 0.001). Approximately one-third (34.9%) of adolescent PCOS patients had characteristic polycystic ovary appearance on ultrasound examination, significantly (*p* = 0.001) more than in the risk and control groups (13.6% and 7.5%, respectively). Overall, acne was more pronounced in the PCOS and risk group than in the control group (*p* < 0.001), with moderate acne being the degree of severity that was significantly more common in PCOS patients (*p* = 0.002). All patients had normal prolactin and thyroid-stimulating hormone levels. Additionally, oestradiol, luteinising hormone and follicle-stimulating hormone levels were in the normal laboratory range of premenopausal women (as per inclusion criteria).

The SNVs’ genotype frequencies are summarized in Table 2. For all six SNVs, there were no significant differences in the genotype frequencies among the PCOS, risk and control groups.

The associations of the PCOS patients’ genotype and phenotypic parameters are shown in Table 3. ESR2 rs4986938 homozygous minor allele carriers had a significantly higher level of T (0.68 ng/mL) than heterozygous individuals (0.38 ng/mL) and homozygous major allele carriers (0.38 ng/mL) (*p* = 0.04). There was a tendency towards a lower mFG score in ESR2 rs4986938 heterozygous individuals; however, this result did not quite reach statistical significance (*p* = 0.056).

The phenotype associations with homozygous major allele carriers (HH) and minor allele carriers (Hh and hh) in the PCOS group are shown in Table 4.

We found that the level of T was significantly higher in LHCGR rs2293275 minor allele carriers (0.61 ng/mL) than homozygous major allele carriers (0.35 ng/mL) (*p* = 0.044). LHCGR rs2293275 minor allele carriers also demonstrated a tendency towards a higher waist-hip ratio compared with homozygous major allele carriers; however, this result did not reach statistical significance (*p* = 0.08). Similarly, although statistical significance was not achieved (*p* = 0.084), FSHR rs6166 minor allele carriers tended towards more severe acne (higher GAGS score) compared with homozygous major allele carriers.

## 4. Discussion

For all six SNVs investigated in the present study—*GNRHR* (rs104893837), *ESR2* (rs4986938), *LHCGR* (rs2293275) and *FSHR* (rs6166, rs6165, rs2349415)—there were no significant differences in the genotype frequencies among the PCOS, risk and control groups. Within the PCOS group, patients who were *ESR2* rs4986938 homozygous minor allele carriers had a significantly higher level of T than heterozygous individuals and homozygous major allele carriers (*p* = 0.04). In addition, the level of T was significantly higher in PCOS patients who were *LHCGR* rs2293275 minor allele carriers than in those who were homozygous major allele carriers (*p* = 0.044). The genotype–phenotype associations of *GNRHR* (rs104893837) and *FSHR* (rs6166, rs6165, rs2349415) in PCOS patients did not reach statistical significance.

The *ESR2* gene is located at chromosome 14q23.2q23.3. It encodes oestrogen receptor 2—a steroid receptor situated in the cell nucleus. *ESR2* rs4986938 is positioned in the non-coding sequence of the gene. Due to *ESR2*’s role in ovarian function, it has been studied in PCOS patients and differences in its ovarian expression level have been reported between patients and controls [9,10]. Several research groups have investigated this SNV’s association with PCOS and its symptoms in different ethnic populations. For instance, Douma and colleagues observed a lower level of hyperandrogenism in Tunisian PCOS patients who were minor allele carriers; however, this association was lost following Bonferroni correction [11]. Indeed, the majority of studies have not found an association between minor allele carrier status and PCOS and its symptoms [11,12,13,14,15]. Nevertheless, Kim et al. found that minor allele carriers had a lower risk of developing PCOS than homozygous major allele carriers in a sample of Korean women [16]. In contrast, Liaqat and co-workers’ study of Pakistani women reported a higher percentage of minor allele carriers among PCOS patients than controls [17]. The mechanism of how this SNV leads to a higher level of testosterone still needs to be elucidated. One suggestion derived from similar results investigating SNV in *ESR1* gene is that the lower expression of ESR could increase the conversion of androgen precursors to testosterone in PCOS women [18].

We found that PCOS patients who were *LHCGR* rs2293275 minor allele carriers had a significantly higher T level than major allele homozygotes. This finding was not affected by the BMI of the participants; however, the association was lost when adjustment for waist-hip ratio was performed. The *LHCGR* gene is located at chromosome 2p16.3. It encodes a receptor for both luteinising hormone and choriogonadotropin hormone. The gene is expressed in different cell types in the ovary, including theca cells and differentiated granulosa cells, and *LHCGR* is thought to transduce luteinising hormone-mediated signals that play a crucial role in the ovulation process [19]. Consequently, if an increased reaction to the circulating level of luteinising hormone occurs, normal follicular development and ovulation are impaired. At the same time, the expression of aromatase increases, facilitating androgen production [20]. The published data regarding rs2293275′s role in the development of PCOS are contradictory. Valkenburg and colleagues did not find an association between this SNV and PCOS development in a European population [21]. However, a meta-analysis conducted by Zou et al. reported a 4.1 risk increase of developing PCOS for carriers of the AA (minor allele) genotype in a Caucasian population [22]. Findings also vary regarding the SNV’s role in the phenotype of PCOS patients. Thathapudi et al.’s study of South Indian women detected an association of the GG (major allele) genotype with BMI, waist-hip ratio, insulin resistance, luteinising hormone level and luteinising hormone/follicle-stimulating hormone ratio in PCOS patients compared with controls. The AA genotype, significant in our study, was linked to a higher basal follicle-stimulating hormone level than non-A allele carriers. The T level was not reported in this particular study [23]. El-Shal et al.’s study of Egyptian PCOS women found an association between *LHCGR* rs2293275 and several anthropometric and biochemical characteristics, including elevated values of the free androgen index and hirsutism score [24]. A study of European PCOS women (The Netherlands) reported a lower basal follicle-stimulating hormone level in patients who had neither the *LHCGR* rs2293275 SNV nor 18insLQ SNV; however, no other clinical associations were detected (14). Another study of a European population (Sardinia) did not find any associations of the SNV with the clinical presentation of PCOS [25].

There were no significant differences in the genotype and allele carrier frequencies of *FSHR* (rs6166, rs6165, rs2349415) among our PCOS, risk and control groups. *FSHR* is a G protein-coupled receptor in the cell membrane of ovarian granulosa cells. Several SNVs in this gene have been described as possible precursors for the development of PCOS. The SNVs rs6166 and rs6165, which result in a change of amino acid in *FSHR* [26], are amongst the most extensively studied variants. Genetic association studies conducted in European populations have reported a link between rs6166 and rs6165 [27] or rs6165 alone [28] and PCOS. Qiu et al. detected a similar association in their candidate gene systematic review of women of European ancestry. They concluded, rs6166 AA (major allele) carriers were less likely to develop PCOS (OR 0.64, 95% CI: 0.42–0.98). However, they did not find an association between rs6165 and the development of PCOS [29]. GWAS performed to date have failed to show statistically significant signals from these SNVs that are related to PCOS [4]. Although both SNVs have been found to be linked to the level of follicle-stimulating hormone in PCOS patients in populations with different ethnic backgrounds, the published data on their role in other clinical and biochemical features of PCOS are contradictory, with the majority showing no association [20,27,29,30]. *FSHR* rs2349415 is located in an intron of the *FSHR* gene. A GWAS performed by Shi et al. found that this SNV was related to PCOS in a Han Chinese population [31]. This link has been confirmed by a meta-analysis of a European population [32] and by a family association study of a Han Chinese population [33].

We did not find a link between *GNRHR* rs104893837 and the development of PCOS or its associated symptoms in our study population. From sequencing of the whole coding part of the *GNRHR* gene, rs104893837 was found to be the most frequent variant in the Latvian population (unpublished results). In humans, GNRH executes its function through the transmembrane G protein-coupled receptor (GNRHR) mainly in the hypophysis but also in ovary and breast tissue. GNRH’s main role is in normal menstrual cycle regulation via stimulation of the luteinising hormone and follicle-stimulating hormone secretion [34] and participation in the formation and atresia of the corpus luteum. One of the main pathophysiological mechanisms underlying the development of PCOS is an increase of the pulse and frequency of GNRH secretion [35]. Several gene association studies have identified numerous SNV associations with the syndrome or with its symptoms [35,36,37,38,39]. Despite the fact that the SNVs we analysed did not show statistically different frequency between PCOS, risk group and control group patients, some alleles were associated with higher T levels in PCOS patients. That could indicate the role of epigenetic factors (e.g., DNA methylation, histone modification) in these alleles in the development of the syndrome and its phenotype. Further research would be necessary to address these aspects.

The differences between the results presented here and previously published ones may be due to different ethnic backgrounds of the studied populations that could influence genetic differences. Furthermore, incongruencies in the diagnostic criteria applied in different studies can influence results. In our study, the PCOS patients were selected using the most recent diagnostic criteria specifically designed for adolescents. Diagnostic criteria have changed significantly over the years, starting with the 1990 NIH criteria [40], then the 2003 Rotterdam criteria [41] and currently the ESHRE 2018 criteria based on rigorous clinical and risk analysis [1]. Therefore, it is very important to use the latest diagnostic criteria with consistency to build a reliable research knowledge base. Re-evaluation of published results to identify at risk patients is very important as these patients may develop symptoms at a later stage of life and ultimately represent a specific subtype of PCOS.

This research has some important strengths and limitations. To the best of our knowledge, this is one of the first genetic studies that have applied the latest ESHRE 2018 criteria. One important change in these diagnostic criteria is that polycystic ovarian morphology is no longer considered as diagnostic criteria in adolescent PCOS patients. It is because relatively large ovaries with numerous follicles is a normal finding in this age group [1]. Additionally, ovary appearance in the ultrasound does not directly correlate with PCOS symptoms, that is also demonstrated by the results in our study where 65.1% of PCOS patients did not have a polycystic ovary morphology in ultrasound, but they still exhibited other PCOS symptoms. The adolescents were recruited from the Children’s Clinical University Hospital in Riga, which is the central children’s hospital in Latvia where most of the country’s paediatric gynaecology services are carried out free of charge and without referral. Therefore, demographic and socioeconomic backgrounds did not influence the selection of participants, thereby providing a study group representative of the whole population. Moreover, all the participants forming the three different groups underwent an identical thorough examination, allowing them to be directly compared with one another. A further major strength of our study is the inclusion of a risk group comprising of patients who have a high chance of developing PCOS at a later stage of life, thus avoiding a skew towards the most severe phenotypes presenting in adolescence. Adolescents in the risk group require close monitoring, and genetic testing is essential to understand the development of particular phenotypes and provide a prognosis for the future course of the disease. A limitation of the study is its small sample size. We intend to test a larger sample and include other SNVs, as well as conduct a follow-up for the current sample (especially the risk patients).

## 5. Conclusions

No statistically significant differences were found in the distributions of genotypes for *GNRHR* (rs104893837), *ESR2* (rs4986938), *LHCGR* (rs2293275) and *FSHR* (rs6166, rs6165, rs2349415) among PCOS patients, risk patients and healthy controls. Within the PCOS group, patients who were *ESR2* rs4986938 homozygous minor allele carriers had a significantly higher level of T than heterozygous individuals and homozygous major allele carriers. In addition, the level of T was significantly higher in PCOS patients who were *LHCGR* rs2293275 minor allele carriers than in those who were homozygous major allele carriers. The genotype-phenotype associations of *GNRHR* (rs104893837) and *FSHR* (rs6166, rs6165, rs2349415) in PCOS patients did not reach statistical significance.

## Figures and Tables

**Table 1 diagnostics-11-02327-t001:** Baseline clinical characteristics of the PCOS, risk and control groups.

Variable	PCOS Group (*n* = 63)	Risk Group (*n* = 22)	Control Group (*n* = 67)	*p*-Value
Gynaecological age, median years (IQR) †	3.0 (2.0)	4.0 (2.0)	4.0 (1.0)	0.441
BMI, median percentile (IQR)	89.9 (48.0)	75.4 (39.8)	55.0 (47.0)	**<0.001** *
Individuals with BMI above the 85th percentile, *n* (%)	31 (49.2)	6 (27.3)	9 (13.4)	**<0.001** **
Waist-hip ratio, median (IQR)	0.82 (0.13)	0.80 (0.06)	0.76 (0.06)	**0.001** *
mFG score, median (IQR)	9.0 (6.0)	8.0 (5.0)	1.0 (2.0)	**<0.001** *
GAGS score, mean (SD)	14.5 (9.1)	10.9 (8.8)	7.0 (6.0)	**<0.001** ^
No acne, *n* (%)	1 (1.9)	2 (10.0)	6 (15.0)	
Mild acne, *n* (%)	35 (64.8)	13 (65.0)	31 (77.5)	0.147 ***
Moderate acne, *n* (%)	16 (29.6)	4 (20.0)	3 (7.5)	**0.002** ***
Severe acne, *n* (%)	2 (3.7)	1 (5.0)	0 (0)	0.091 ***
Polycystic ovary morphology on ultrasound, *n* (%)	22 (34.9)	3 (13.6)	5 (7.5)	**0.001** ***

† Gynaecological age—age (in years) at the time of the study minus age at menarche; BMI—body mass index; mFG—modified Ferriman-Gallwey; GAGS—Global Acne Grading System; * Kruskal-Wallis test; ** Pearson’s chi-squared test; *** Fisher’s exact test; ^ one-way ANOVA; IQR—interquartile range for medians; statistically significant *p*-values are denoted in bold.

**Table 2 diagnostics-11-02327-t002:** Genotype frequencies of the six SNVs in the PCOS, risk and control groups.

SNV/Genotype	PCOS Group (*n* = 63)			Risk Group (*n* = 22)			Control Group (*n* = 67)			*p*-Value
	HH *n* (%)	Hh *n* (%)	hh *n* (%)	HH *n* (%)	Hh *n* (%)	hh *n* (%)	HH *n* (%)	Hh *n* (%)	hh *n* (%)	
*FSHR* rs2349415	25 (39.7)	28 (44.4)	10 (15.9)	7 (31.8)	11 (50.0)	4 (18.2)	20 (29.9)	33 (49.3)	14 (20.9)	0.81 *
*FSHR* rs6166	18 (28.6)	30 (47.6)	15 (23.8)	7 (31.8)	12 (54.5)	3 (13.6)	24 (35.8)	29 (43.3)	14 (20.9)	0.77 *
*FSHR* rs6165	15 (23.8)	36 (57.1)	12 (19.1)	5 (22.7)	13 (59.1)	4 (18.2)	21 (31.3)	31 (46.3)	15 (22.4)	0.73 *
*ESR2* rs4986938	14 (22.2)	38 (60.3)	11 (17.5)	5 (22.7)	13 (59.1)	4 (18.2)	10 (14.9)	49 (73.1)	8 (11.9)	0.58 *
*GNRHR* rs104893837	59 (93.7)	4 (6.3)	0 (0)	21 (95.5)	1 (4.5)	0 (0)	64 (95.5)	3 (4.5)	0 (0)	0.89 **
*LHCGR* rs2293275	26 (41.3)	26 (41.3)	11 (17.4)	6 (27.3)	11 (50.0)	5 (22.7)	22 (32.8)	31 (46.3)	14 (20.9)	0.77 *

* Pearson’s chi-squared test; ** Fisher’s exact test; HH—homozygous carriers of major alleles; Hh—heterozygous allele carriers; hh—homozygous carriers of minor alleles.

**Table 3 diagnostics-11-02327-t003:** Genotype-phenotype associations in PCOS patients.

SNV	Modified Ferriman-Gallwey Score, Median (IQR)				BMI Percentile, Median (IQR)				Waist-Hip Ratio, Median (IQR)				Total Testosterone Level, Median (IQR)				GAGS Score, Mean (SD)				PCO Morphology on Ultrasound, *n* (%)			
	HH	Hh	hh	*p* *	HH	Hh	hh	*p* *	HH	Hh	hh	*p* *	HH	Hh	hh	*p* *	HH	Hh	hh	*p* ^	HH	Hh	hh	*p*
*FSHR* rs2349415	10.0 (6.0)	9.0 (6.0)	12.5 (9.8)	0.44	82.2 (64.1)	93.1 (45.0)	91.4 (26.1)	0.88	0.84 (0.19)	0.84 (0.11)	0.77 (0.17)	0.65	0.42 (0.37)	0.48 (0.52)	0.40 (0.37)	0.76	15.0 (10.7)	13.8 (7.6)	13.5 (8.1)	0.99	9 (40.9)	10 (45.5)	3 (13.6)	0.849 **
*FSHR* rs6166	9.0 (7.5)	9.5 (6.0)	14.0 (10.5)	0.19	95.6 (60.6)	76.7 (46.6)	81.8 (60.4)	0.93	0.83 (0.17)	0.82 (0.13)	0.84 (0.10)	0.89	0.53 (0.32)	0.38 (0.44)	0.44 (0.42)	0.80	10.4 (7.8)	16.5 (9.4)	14.0 (8.4)	0.12	5 (22.7)	10 (45.5)	7 (31.8)	0.371 **
*FSHR* rs6165	9.0 (7.0)	9.0 (6.0)	15.0 (8.0)	0.12	95.6 (63.9)	76.7 (48.0)	94.0 (41.4)	0.71	0.84 (0.16)	0.82 (0.14)	0.84 (0.13)	0.94	0.45 (0.49)	0.40 (0.41)	0.57 (0.43)	0.89	12.4 (8.8)	15.1 (9.3)	13.4 (9.2)	0.70	4 (18.2)	12 (54.5)	6 (27.3)	0.492 ***
*ESR2* rs4986938	10.0 (6.0)	8.0 (7.3)	11.0 (6.5)	0.056	90.4 (49.9)	89.0 (53.9)	96.6 (30.8)	0.36	0.81 (0.09)	0.81 (0.14)	0.85 (0.05)	0.63	0.38 (0.26)	0.38 (0.38)	0.68 (0.20)	**0.04**	14.0 (10.9)	13.3 (15.3)	13.4 (16.5)	0.71	6 (27.3)	13 (59.1)	3 (13.6)	0.791 ***
*GNRHR* rs104893837	9.5 (6.0)	10.0	0 (0)	0.99	90.4 (48.0)	74.9	0 (0)	0.75	0.83 (0.13)	0.88	0 (0)	0.97	0.44 (0.40)	0.33	0 (0)	0.47	14.4 (8.9)	12.5 (16.3)	0 (0)	0.31	21 (95.5)	1 (4.5)	0 (0)	1.000 ***
*LHCGR* rs2293275	9.0 (6.0)	10.5 (5.8)	14.0 (12.0)	0.92	71.6 (56.8)	89.0 (49.4)	98.9 (15.9)	0.27	0.81 (0.11)	0.84 (0.17)	0.88 (0.09)	0.19	0.35 (0.37)	0.59 (0.36)	0.67 (0.66)	0.13	14.3 (9.0)	15.6 (9.6)	10.6 (7.6)	0.60	9 (40.9)	9 (40.9)	4 (18.2)	0.814 ***

BMI—body mass index; GAGS—Global Acne Grading System; PCO—polycystic ovary; HH—homozygous carriers of major alleles; Hh—heterozygous allele carriers; hh—homozygous carriers of minor alleles; * Kruskal-Wallis test; ** Pearson’s chi-squared test; *** Fisher’s exact test ^ one-way ANOVA; IQR—interquartile range for medians; statistically significant *p*-value is denoted in bold.

**Table 4 diagnostics-11-02327-t004:** Phenotype associations with homozygous major (HH) and minor (Hh/hh) allele carriers in the PCOS group.

SNV	Modified Ferriman-Gallwey Score, Median (IQR)			BMI Percentile, Median (IQR)			Waist-Hip Ratio, Median (IQR)			Total Testosterone Level, Median (IQR)			GAGS Score, Mean (SD)			PCO Morphology on Ultrasound, *n* (%)		
	Major Allele Homozygotes (HH)	Minor Allele Carriers (Hh, hh)	*p* *	Major Allele Homozygotes (HH)	Minor Allele Carriers (Hh, hh)	*p* *	Major Allele Homozygotes (HH)	Minor Allele Carriers (Hh, hh)	*p* *	Major Allele Homozygotes (HH)	Minor Allele Carriers (Hh, hh)	*p* *	Major Allele Homozygotes (HH)	Minor Allele Carriers (Hh, hh)	*p* ^	Major Allele Homozygotes (HH)	Minor Allele Carriers (Hh, hh)	*p*
*FSHR* rs2349415	10.0 (6.0)	9.5 (7.8)	0.84	82.2 (64.1)	92.3 (41.2)	0.73	0.84 (0.19)	0.83 (0.12)	0.36	0.42 (0.37)	0.46 (0.47)	0.90	14.9 (10.7)	14.0 (7.6)	0.902	9 (40.9)	13 (59.1)	0.879 **
*FSHR* rs6166	9.0 (7.5)	10.0 (7.0)	0.60	95.6 (60.6)	76.7 (48.1)	0.73	0.83 (0.17)	0.84 (0.13)	0.90	0.53 (0.32)	0.38 (0.42)	0.50	10.4 (7.8)	16.3 (9.1)	0.084	5 (22.7)	17 (77.3)	0.389 **
*FSHR* rs6165	9.0 (7.0)	10.0 (6.0)	0.64	95.6 (63.9)	83.7 (48.1)	0.79	0.84 (0.16)	0.83 (0.14)	0.98	0.45 (0.49)	0.43 (0.39)	0.74	12.4 (8.6)	15.1 (9.2)	0.434	4 (18.2)	18 (81.8)	0.747 ***
*ESR2* rs4986938	10.0 (6.0)	8.0 (7.0)	0.07	90.4 (49.9)	89.4 (47.1)	0.92	0.81 (0.09)	0.83 (0.12)	0.45	0.38 (0.26)	0.47 (0.42)	0.40	14.0 (10.9)	13.3 (8.7)	0.476	6 (27.3)	16 (72.7)	0.566 ***
*GNRHR* rs104893837	10.0 (6.0)	10.0	0.99	90.4 (48.0)	74.9	0.75	0.84 (0.12)	0.88	0.97	0.45 (0.40)	0.33	0.47	14.5 (9.0)	12.5 (16.3)	0.312	21 (95.5)	1 (4.5)	1.000 ***
*LHCGR* rs2293275	9.0 (6.0)	11.5 (6.3)	0.71	71.6 (56.8)	94.4 (44.5)	0.37	0.81 (0.11)	0.86 (0.11)	0.08	0.35 (0.37)	0.61 (0.42)	**0.044**	14.3 (9.0)	14.5 (9.4)	0.902	9 (40.9)	13 (59.1)	0.792 **

BMI—body mass index; GAGS—Global Acne Grading System; PCO—polycystic ovary; HH—homozygous carriers of major alleles; Hh—heterozygous allele carriers; hh—homozygous carriers of minor alleles; * Kruskal-Wallis test; ** Pearson’s chi-squared test; *** Fisher’s exact test; ^ Student’s *t*-test; IQR—interquartile range for medians; statistically significant *p*-value is denoted in bold.

## Data Availability

Data available on request due to restrictions.

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
