# Peer review of "Role of Single Nucleotide Variants in FSHR, GNRHR, ESR2 and LHCGR Genes in Adolescents with Polycystic Ovary Syndrome"

_diagnostics, 2021, doi:10.3390/diagnostics11122327_

Round 1

Reviewer 1 Report

The manuscript described the distribution of single nucleotide variants (SNVs) in FSHR, GNRHR, ESR2 and LHCGR genes among adlescent PCOS patients and control subjects. The authors found no statsitical differences in the distribution of SNVs of these genes among PCOS patients, risk patients and control subjects. And, importantly they found that in ESR2 minor allele homozygous (hh) patients  higher testosterone levels were detected. The ESR2 SNV (re49869389) is located in the non-coding region of ESR2 gene. Hence, estrogen receptor functions of the ESR2 SNV seem to be not affected. ESR2 expression in the ovary in the minor allele homozygous patients may be varied. These results may suggests possible involvement of estrogen receptor (ESR2) on the development of PCOS or testosterone production, although possible mechanism has been not discussed. The present study seems to have merits for publication for understanding PCOS. However, some points shown below may need to be considered.

 1.  I want to know ovarian structures of ESR2 minor allele homozygous patients  with higher testosterone levels. In another words, hh patients with non-PCO ovaries showed higher testosterone levels or not. Correlation between structural changes of ovaries and functional charges in these PCOS patients may be discussed.

2. Line 169; authors may discuss reason why the adolescent PCOS patients with non-PCO showd PCOS symptoms.

3. Possible expalnation on the higher testosterone levels in the hh patients of ERS2 SNV may be needed.

Author Response

Thank you for editing our manuscript! We found your suggestions very useful and tried to implement those as best as we could. We sincerely hope this has added more clarity and value to the manuscript.

Reviewer No1.

  1. Thank you for your thorough analysis. We indeed found statistically significant differences in the level of testosterone among hh genotype carriers in ESR2 with PCOM in ultrasound and without that (p=0.039). Nevertheless, as the patient number was verry small in both groups (n=4 with PCOM and n=7 without it), we felt that we should increase the number of patients for us to feel comfortable in presenting these results as indeed statistically significant.
  2. We have addressed this issue in the discussion section. We hope this addition clarifies the issue.

  3. We have addressed this issue in the discussion section. We hope this addition clarifies the issue.

Reviewer 2 Report

Lidaka and associates have submitted an interesting report on frequency of selected nucleotide variants across three populations in Latvia with reference to PCOS. Although no differences were identified among the 4 markers chosen for study, there were differences in serum testosterone (higher among PCOS cases). This research adds to the literature describing PCOS and should be considered for publication, after the authors address a few points:

Major concern

Based on these data, genetic screening for PCOS using these variants would have little to no clinical value. These data show zero predictive or associative relation with these alleles & PCOS. However, since androgen levels were significantly higher in PCOS as a function of various alleles, this suggests epigenetic/other factors are important in driving PCOS pathology.

Do the authors believe they have done enough in the discussion section to explain why this ‘negative’ result nevertheless supports further genetic/epigenetic investigation regarding PCOS? If their efforts failed to reveal any genomic link, then what will?

The article refers to Thathapudi et al (line 254-257) where PCOS metabolic testing was performed. But in the current research, neither fasting insulin nor 3h IGTT were done to assess future risk of PCOS in these Latvian patients. Why not?

Minor items

At line 57, here the authors may wish to rephrase as ‘more than 15%’

At line 60: The main treatment for PCOS is not always ‘lifestyle changes’. For example, in USA the top treatment for PCOS is oral contraceptive pills prescribed to re-establish menstrual regularity. As this publication is likely to receive wide readership, it would be wise to avoid statements about treatments which might be common locally, but not widely used elsewhere.

Line 83: suggest ‘because’ rather than ‘as’ here.

Introducing the term ‘cross-sectional’ (line 90) is good, provided the general background population from which the sample was drawn is carefully described. This last part is underdeveloped here, so the authors may wish to rework the explanation to clarify their denominator somewhat.

Line 127:  Since PCOS patients often have absent/irregular cycles, how was sample collection between cycle day 3-5 actually scheduled? In other words, if some women had no cycles at all, then one may assume that no samples were supplied from such patients with amenorrhea. Is that correct?

Line 162: ‘Gynecological age’ is an odd term to invent for a basic concept. It appears to mean ‘years since menarche’ .. If yes, just say so.

Line 300: Emphasizing ethic differences is commendable, but since this work benefits from a Latvian registry where a homogenous (mono-ethnic) population was studied, this becomes an important asset for the index investigation. Perhaps the title should indicate this?

Line 306: split infinitive noted, suggest ‘…it is important to use latest diagnostic criteria with consistency …’

Author Response

1)Thank you for a very valuable point! We included your considerations in discussion section. We definitely agree that more research is needed in this direction.

2) Yes, we agree that screening of metabolic changes would add a value to this research. There were several reasons why those were not added in this publication. Firstly, as PCOS encompasses wide range of aspects, to do more focused analysis, we analysed diagnostic criteria that are included in ESHRE, 2018 guidelines. Secondly, as the sample were still adolescents, we saw that significant metabolic changes were present only in a small number of patients, decreasing the chance to reach a statistical power of these results. Nevertheless, we are planning to do a follow-up for this sample.

Minor items:

  • Line 57 – corrected.
  • Line 60 - Explanation added.
  • Line 83 – corrected.
  • Term cross-sectional.
  • We added some additional information about the recruitment facility and homogeneity of the sample. We thought that maybe the change of the title would be too ambitious, but we hope that these explanations give a clearer view on a study sample.
  • Thank you for the valuable point of patients with amenorrhea. We added the explanation how we tried to avoid false results taken in periovulatory or lutheal phase of menstrual cycle.
  • We added the explanation regarding gynaecological age. That is a term used by ESHRE, 2018 guidelines, thus we felt that we need to include this in our publication as well. Hope it is less confusing now.
  • Line 306 – corrected.